# Recruitment and Baseline Characteristics of Participants in the “Sanadak” Trial: A Self-Help App for Syrian Refugees with Post-traumatic Stress

**DOI:** 10.3390/ijerph17207578

**Published:** 2020-10-18

**Authors:** Susanne Röhr, Franziska U. Jung, Anna Renner, Anna Plexnies, Rahel Hoffmann, Judith Dams, Thomas Grochtdreis, Hans-Helmut König, Anette Kersting, Steffi G. Riedel-Heller

**Affiliations:** 1Institute of Social Medicine, Occupational Health and Public Health (ISAP), Medical Faculty, University of Leipzig, 04103 Leipzig, Germany; franziska.jung@medizin.uni-leipzig.de (F.U.J.); steffi.riedel-heller@medizin.uni-leipzig.de (S.G.R.-H.); 2Global Brain Health Institute (GBHI), Trinity College Dublin, D02 PN40 Dublin, Ireland; 3Department of Psychosomatic Medicine and Psychotherapy, University Medical Center Leipzig, 04103 Leipzig, Germany; anna.renner@medizin.uni-leipzig.de (A.R.); pt.plexnies@gmail.com (A.P.); rahelhoffmann@posteo.de (R.H.); anette.kersting@medizin.uni-leipzig.de (A.K.); 4Department of Health Economics and Health Services Research, Hamburg Center for Health Economics, University Medical Center Hamburg-Eppendorf, 20246 Hamburg, Germany; j.dams@uke.de (J.D.); t.grochtdreis@uke.de (T.G.); h.koenig@uke.de (H-H.K.)

**Keywords:** Syrian refugees, posttraumatic stress, eHealth, mHealth, app, smartphone, trial, randomized-controlled trial, intervention, baseline, mental health

## Abstract

Many Syrian refugees residing in Germany have been exposed to traumatizing events, while treatment options are scarce. Therefore, the self-help app “Sanadak” was developed to target post-traumatic stress in Syrian refugees. We aimed to inspect the recruitment and baseline characteristics of the participants in the trial, which is conducted to evaluate the app. Analyses were based on the recruitment sample (*n* = 170) and the trial sample (*n* = 133). Data were collected during structured face-to-face interviews in the Arabic language. Targeted outcomes included post-traumatic stress (primary; Post-traumatic Diagnostic Scale for DSM-5/PDS-5) and depressive symptoms, anxiety, resilience, among others (secondary). Recruited individuals were *M* = 32.8 (SD = 11.2, range = 18–65) years old; 38.8% were women. The average PDS-5 score was 23.6 (SD = 13.2) regarding trauma exposure, which was most frequently related to experiencing military- or combat-related events (32.9%). Moreover, 46.5% had major depression and 51.8% showed low resilience. Anxiety was present in 40.6% of the trial participants. Psychological distress was high in Syrian refugees residing in Germany, enrolled in a trial targeting post-traumatic stress. This underlines the need for intervention. Our results provide important figures on the mental health of a not well-studied population group in Germany.

## 1. Introduction

About 790,000 Syrian refugees have arrived in Germany to take shelter since the years 2010/2011, which marked the eruption of the Syrian civil war [1]. The majority of Syrian refugees were exposed to a variety of potentially traumatizing events, such as military combat, torture or imprisonment. Regardless whether such events were witnessed or personally experienced, they may increase vulnerability to post-traumatic stress and related adverse mental health outcomes [2]. A study reported that 75.3% among a group of 518 adult Syrian refugees in Germany had witnessed and/or experienced traumatic events. Subsequently, symptoms of post-traumatic stress disorder (PTSD) were reported by 11.4% of them [2]. Additionally, PTSD is often associated with a range of comorbidities that further compromise mental health. The most common comorbidities are: mood disorders (such as depression or bipolar disorders), anxiety disorders (such as generalized anxiety disorder, panic disorder), substance dependence, somatization disorders and increased attempts of suicide [3,4,5]. For example, moderate to severe depression was present in 14.5% and mild to severe generalized anxiety in 13.5% among Syrian adult refugees [2]. PTSD can furthermore result in distress or impaired social functioning [6,7,8]. Traumatic experiences tend to be stored in the implicit memory, but often only in parts, which makes a holistic narrative impossible. This frequently leads to intrusions and avoidance behaviour. In addition to post-traumatic stress, problems with residential status in the host country can further negatively impact mental health [9]. Therefore, psychotherapeutic interventions are highly indicated, and it has been recommended that they should be offered promptly after the arrival of refugees [10,11]. If treatment is not available, symptoms of PTSD and depression may be present in refugees even 20 years after their escape [12].

Indeed, there is a lack of adequate treatment possibilities due to a lack of sufficient treatment centres that offer psychological help or psychotherapy for refugees in host countries. This is mainly due to intercultural and language barriers that hinder utilization of help. Therefore, e-health options such as mobile phone-based interventions may be a possibility to fill the gap. This was the aim of our project Help@App. We developed a low-threshold self-help app “Sanadak” in Arabic language that is built on evidence-based cognitive-behavioural therapy (CBT) for PTSD. The content of the “Sanadak” app is multi-modal, i.e., it includes psychoeducational information to increase knowledge and awareness of PTSD and related mental health issues, and self-help techniques as well as skills training with respect to symptom management. In addition, a short self-test on post-traumatic symptom severity is implemented to allow for automated tailored feedback regarding progress at any time. Interactive materials, such as animated videos and audios as well as games and exercises are provided to maximize usability. The effectiveness of the app is evaluated in a randomized-controlled trial (RCT) with two arms: the intervention group uses the app and the control group receives psychoeducational reading material in order to increase knowledge and awareness of PTSD. Details have been described elsewhere [13].

The aim of this report was to describe (1) the recruitment processes including details on drop-out and eligibility for randomization as well as recruitment sample characteristics and (2) baseline characteristics of study participants with regard to group allocation (intervention and control group).

## 2. Materials and Methods

### 2.1. Recruitment and Recruitment Procedures

Recruitment and eligibility screening began in October 2018 in Leipzig, Germany. Participants were recruited in a multi-modal manner, as described elsewhere in detail [14]. Recruitment methods included, among others, snowball sampling, personal contacts of the study personnel who were conducting recruitment and interviews as well as cooperation with multipliers, i.e., facilities that actively work with the target group. Comprehensive study material in Arabic language was developed and used to further attract new prospects and to inform about the study. In addition, a broad range of contact possibilities (i.e., WhatsApp, Telegram, email, telephone contact, and social media) was offered.

Individuals who signaled interest in taking part in the study were screened to assess eligibility for trial participation with regard to post-traumatic stress symptom severity, symptoms of depression, and suicidal risk in accordance with the study protocol [13]. Inclusion criteria were: Syrian refugee living in Germany, aged 18–65 years, the experience of at least one traumatic event and subsequent mild to moderate post-traumatic stress symptom severity (Posttraumatic Diagnostic Scale for DSM-5, PDS-5 = 11–59 [15]) and owning a compatible device in order to be able to use the app (Android/iOS). Moreover, literacy was a requirement. Exclusion criteria included post-traumatic stress symptomatology outside of the range mentioned above, severe depressive symptoms (Patient Health Questionnaire, PHQ-9 ≥ 20 [16]), acute suicidal tendencies (Depressive Symptom Inventory-Suicidality Subscale, DSI-SS ≥3 [17]), current psychotherapy/psychiatric treatment and/or psychotropic medication as well as pregnancy. If individuals were not eligible for trial participation due to severity of symptoms, they received psychoeducational material on mental health care and contact information of regional initiatives that offer face-to-face support. In five cases, the DSI-SS suicidal score was slightly over the pre-defined cutoff score of 3. Eligibility was still considered based on the overall clinical impression of each prospect; as all of them credibly demonstrated no intention to act. These decisions were reached in consensus conferences between study nurses and study psychologists. Stability in inclusion and exclusion criteria were monitored during baseline- and follow up-assessments to ensure compliance with the study protocol. The recruitment process and sample selection are summarized in Figure 1. Recruitment and eligibility screening processes were completed in December 2019.

### 2.2. Randomization

Participants were randomly allocated to the intervention or control group using a 1:1 ratio utilizing randomized permuted blocks of six, stratified by age and sex in order to ensure balance in sample size and distribution of covariates. The randomization block list was generated by an external, independent statistician using the “blockrand” package provided by the statistics software R. These block lists were then coded, leaving the person who was responsible for group allocation blind to the strata identity.

### 2.3. Assessments and Instruments

Overall, the study included up to four assessments (screening, baseline, follow-up 1 and 2). During each assessment, trained study nurses interviewed a participant using a written, structured questionnaire (paper-and-pencil assessment, face-to-face). The first assessment was the eligibility screening, outlined above, including questions on sociodemographic characteristics, information related to the escape, and instruments for measuring eligibility in accordance to mental health. In addition to the measures described in the *Recruitment* section above, resilience was assessed using the Resilience Scale (RS-13, [18]) during screening.

Sociodemographic characteristics included information on age, gender, net personal income, living situation, residence status, religious group and religiosity (Centrality of Religiosity Scale/CRS, [19]). Information on education was gathered based on scholastic and professional qualifications with regard to the Syrian educational system. We then categorized the level of education (low, medium, high) based on the Comparative Analysis of Social Mobility in Industrial Nations (CASMIN) educational classification system [20].

After the screening, eligible individuals, who agreed to take part in the trial, underwent an additional comprehensive assessment on psychosocial health and associated factors (baseline assessment), including, among others, measures on generalized anxiety (GAD-7, [21]), severity of somatic symptoms (PHQ-15, [22]), general self-efficacy (GSE, [23]); self-stigma (Self-stigma of Mental Illness Scale–Short Form, SSMIS-SF, [24]); social support (short form of the Lubben Social Network Scale, LSNS-6, [25]; ENRICHD Social Support Inventory, ESSI, [26]); and health-related quality of life and health status (5-level version of EQ-5D and EQVAS, EQ-5D-5L, [27]). Please see the aforementioned study protocol for further information [13].

### 2.4. Data Entry and Data Quality Control

Data entry took place immediately after data collection using the statistical software SPSS 24 (IBM Corp., Armonk, NY, USA). Concurrently, data completeness and consistency checks were conducted to ensure data integrity. With regard to escape-related information, three participants refused to give answers. In addition, another three participants refused to give an answer on the item asessing “pain or problems during sexual intercourse” of the PHQ-15. Missing values were replaced with the item’s mean score of all available responses [28]. The database is stored locally and only study personnel who have signed data protection wavers are able to access it. Each participant was given an ID; therefore, pseudonymity was ensured.

### 2.5. Ethics and Registration

The study was approved by the Ethics committee of the Medical Faculty of the University of Leipzig, Germany (ID: 111–17-ek) and adheres to the Declaration of Helsinki. All participants were informed about the study aims, including clarification about data security according to latest legal standards. Participation was only allowed after written informed consent. The study was pre-registered at the German Clinical Trials Register/Deutsches Register Klinischer Studien (DRKS; registration ID: DRKS00013782; date: 6th of July 2018). 

### 2.6. Statistical Analyses

Group differences in the recruitment sample (randomized vs. nonrandomized) and in the study sample (intervention group vs. control group) were inspected using Chi-squared tests and *t*-tests, as appropriate. Analyses were conducted using Stata 16 (SE; StataCorp., College Station, TX, USA).

## 3. Results

### 3.1. Recruitment Sample Characteristics

Results with regard to the recruitment sample are summarized in Table 1 (sociodemographic characteristics), Table 2 (escape-related information, traumatic events and post-traumatic stress symptoms) and Table 3 (secondary outcomes).

#### 3.1.1. Sociodemographic Characteristics of the Recruitment Sample

The mean age of the screened participants was 32.8 (*SD* = 11.2) years and the majority was male (61.2%) (Table 1). Overall, 66.5% went to school for 12 or more years. In terms of net personal income, more than 73% of participants indicated that their current income was less than 1000 Euros per month. Concerning the living situation and family status, 57.6% of participants were currently living with family members or relatives; 48.8% were single and 40.6% were married. Regarding their residence status, 12.3% of the screening participants were “asylum seeker”, 43.5% were entitled to asylum and 23.5% fell under the “refugee protection status”. Though the majority had a working permit (88.8%), about two-thirds reported to be unemployed (68.8%); however, 35.0% of those were students.

Five eligible individuals refused further participation after randomization.

#### 3.1.2. Escape- and Trauma-Related Characteristics of the Recruitment Sample

Findings with regard to escape-related information, traumatic events and post-traumatic stress symptoms are summarized in Table 2. The majority of participants escaped via land (66.5%) and sea (58.2%); followed by airplanes (44.7%), and 18.2% stated that they escaped via transit countries. On average, participants had left Syria more than four years ago (49.8 months) and have spent about three and a half years (41 months) in Germany. 

Most frequently experienced traumatic events were military or combat-related events (32.9%) and “others” (38.8%), which were mostly specified as escape-related events. In two cases, randomized participants were not willing to specify their most stressful traumatic event. However, they were referring to the nonspecified event when answering questions on the PDS-5-scale. The total PDS5-score was *M* = 23.6 (*SD* = 13.2; randomized group: *M* = 24.4, *SD* = 11.1; nonrandomized group: *M* = 20.3, *SD* = 20.5; *p* = 133). The five participants that dropped out after being randomized had a mean PDS-5-score of 23.8 (*SD* = 8.5).

#### 3.1.3. Secondary Mental Health Characteristics of the Recruitment Sample

Secondary outcomes, including depressive symptoms, resilience and suicidal risk are summarized in Table 3. The overall PHQ-9 score indicating depressive symptoms was 9.4 (*SD* = 5.8) on average. Assuming major depression with a cut-off value of 10 or more, 46.5% of the recruitment sample had major depression (47.7% in the randomized group and 40.6% in the nonrandomized group; *p* = 0.492) [16].

The resilience score, measured using the RS-13, was 64.3 (*SD* = 12.7) on average, which is considered low resilience. Results of the Depressive Symptom Inventory-Suicidality Subscale (DSI-SS) showed a mean score of 0.4 (*SD* = 1.3), with 0.2 (*SD* = 1.0) in the randomized group and 1.0 (*SD* = 2.1) in the nonrandomized group (*p* < 0.05). Higher scores in the nonrandomized group are due to the exclusion of elevated suicidal risk.

### 3.2. Study Sample Characteristics

After successful screening, 133 study participants were randomized into the intervention group (*n* = 65) or control group (*n* = 68). Results with regard to the study sample are described in Table 4 (sociodemographic characteristics), Table 5 (escape-related information, traumatic events and posttraumatic stress symptoms) and Table 6 (secondary outcomes). There were no significant group differences with regard to sociodemographic characteristics. 

#### 3.2.1. Escape and Trauma-Related Characteristics of the “Sanadak” Study Sample

Details on escape- and trauma-related characteristics are summarized in Table 5. Again, there were no significant differences between the intervention group and the control group.

#### 3.2.2. Secondary Mental Health Characteristics of the “Sanadak” Study Sample

Secondary outcome measures at baseline are described in Table 6. Significant group differences between the intervention and control groups were found in regard to resilience and self-stigmatization. The overall RS-13 score was higher in the control group compared to the intervention group (*M* = 67.0, *SD* = 10.3 vs. *M* = 61.6, *SD* = 11.6; *p* < 0.05). In relation to stigmatization (SSMIS-SF scale), the control group showed significantly higher scores with regard to the subscale *agreement* (*M* = 20.0, *SD* = 6.8 vs. *M* = 17.6, *SD* = 6.5; *p* < 0.05). No group differences were found for other targeted variables.

## 4. Discussion

The aim of the project Help@App is to develop and evaluate the effectiveness of the interactive, Arabic language self-help app “Sanadak”, which targets Syrian refugees with post-traumatic stress. Based on the results of the screening and baseline assessments, this report provides useful information before engaging in the RCT.

The main reason for noneligibility for trial participation after the screening was that the severity of post-traumatic stress symptoms did not meet the study’s inclusion criteria. Out of 138 eligible individuals being randomized, five decided not to take part in the study and could therefore not be scheduled for baseline assessments. Finally, 133 individuals were allocated to the intervention group (*n* = 65) and to the control group (*n* = 68) and constituted the study sample of the “Sanadak” trial. Importantly, the randomization strategy proved to be successful as there were no group differences with regard to key sociodemographic and primary mental health outcomes after allocating eligible participants to the intervention group or control group. Furthermore, the presented results provide important key figures of a large, but not well-studied population group in Germany.

Previous studies investigating post-traumatic stress in refugees relied on a variety of different measures. Therefore, it is difficult to compare our results to other studies with a similar focus. The original study, that investigated the psychometric properties of the PDS-5 relied on a broad sample of individuals with different ethnical backgrounds that had experienced a traumatic event according to the DSM-5 criteria: the authors reported higher post-traumatic stress total scores in their sample compared to ours [15]. Further differences between the samples related to the type of traumas. Whereas physical assault, for example, was reported more frequently in the sample of Foa et al. [15] (21.2% vs. 6.5%), military or combat-related trauma was much more often reported in our sample of Syrian refugees (12.6% vs. 32.9%).

In addition to post-traumatic stress, previous work on mental health in refugees and migrants mostly focused on depression and anxiety. Regarding depression, the majority of Syrian refugees reported increased depressive symptoms and over 46% of our recruitment sample had major depression. In a (nonrepresentative) survey with over 500 community-dwelling adult Syrian refugees, the PHQ-9 depression mean score was slightly lower (7.1 compared to 9.2 in our study sample) [2]. However, a study with Syrian refugees living in housing facilities in Sweden reported a higher mean score (11.5) [30]. In an online survey by Euteneuer et al. [31], that comprised male Syrian refugees residing in Germany recruited via refugee information centers, the mean score was also higher (10.4) and 29% of the refugees classified for probable major depression. The aforementioned study by Georgiadou et al. [2] found moderate to severe generalized anxiety in 13.5% among their sample of community-dwelling adult Syrian refugees compared to over 40% in ours. Further analysis of somatic symptoms revealed higher symptom severity in female participants compared to male participants (PHQ-15 mean scores: 10.5 vs. 7.8), which was in line with previous study results [11], confirming gender-specific differences in somatisation in refugees. The number of individuals with acute suicidal tendencies in our sample was low. This is contrary to other studies that reported higher suicidality in refugees, including a population-based cohort [32] and compared to the general population [33,34,35]. However, as already stated before, the concept of suicidality may have lead to misunderstandings in our sample due to cultural differences. With regard to general self-efficacy, the mean score in our study sample was in good agreement with a study including Syrian participants (not described as refugees) [36]. We could not identify studies that adapted the resilience scale, RS-13, therefore, scores cannot be compared. However, with regard to the general population in Germany [18], the mean score in our sample was lower (64.3 vs. 70.0), therefore, the participants in our sample could be considered less resilient. However, our sample comprises individuals with relevant post-traumatic stress and high global psychological distress, which may be a result of lower resilience in the first place. The role of resilience requires further study. A significant difference was found with regard to the resilience mean scores in the intervention and control group. Participants allocated to the control group scored higher on the resilience scale. This difference will be addressed in the trial outcome analysis.

Social support, which has been described as a potential protective factor against PTSD, has also been a focus of this study. Less than half of our study participants reported high social support, which is comparable to a study that investigated social support in a random sample of Syrian refugees residing in Sweden [37]. Lastly, scores on self-stigma in relation to mental illness were in line with other studies, that used the same measure but focused on participants with serious mental health complaints without a refugee background [24].

## 5. Limitations

With regard to depressive symptoms as well as post-traumatic stress, we found slight deviations between the screening and the recruitment sample in regard to the randomized group (for PHQ-9: the mean score changed from 9.4 to 9.2; for PDS-5 the score changed from 24.4 to 23.8). This can be explained by the study procedure: In the majority of cases, baseline assessments were scheduled shortly after the screening assessments. However, in few cases, the time period between screening and baseline assessment exceeded two or more weeks due to difficulties in scheduling interviews with the participants, which required repeating the assessment of the PHQ-9 and PDS-5 in order to measure current states. Moreover, result comparisons with other studies investigating mental and social health outcomes in Syrian refugees have to be viewed with caution as sample selection tended to vary and samples were usually nonrandom. Lastly, even though assessments were very comprehensive (average duration: 2.5 to 3 h), there may be other relevant factors, for example, perceived discrimination in the host country, that we were not able to cover due to time constraints.

## 6. Conclusions

The results of the recruitment and randomization procedures proved the allocation strategy to be successful, since distinctive characteristics with regard to sociodemographic information did not vary between targeted groups. Furthermore, the results of this report provide an important basis for evaluating the “Sanadak” app and can also provide reference points for other studies with similar aims. Overall, participants of the Help@App study showed high global psychological distress across considered outcomes, which highlights the urgent need for intervention.

## Figures and Tables

**Figure 1 ijerph-17-07578-f001:**
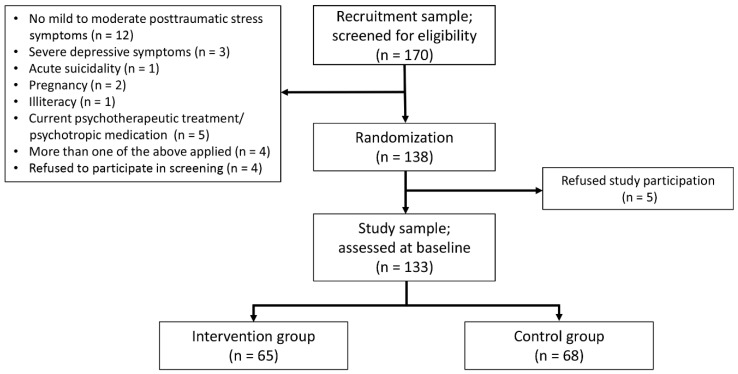
Flow chart of the recruitment process and sample selection.

**Table 1 ijerph-17-07578-t001:** Sociodemographic characteristics of the recruitment sample screened for eligibility in the “Sanadak” trial with regard to randomization status.

Variables	Recruitment Sample	Randomized ^1^	Non-Randomized ^1^	*p*-Value
	(*n* = 170)	(*n* = 133)	(*n* = 32)	
Age (M, SD)	Ø 32.8 (11.2)	Ø 33.3 (11.2)	Ø 30.4 (11.8)	0.19
Gender				
Male	104 (61.2%)	82 (61.6%)	19 (59.4%)	
Female	66 (38.8%)	51 (38.3%)	13 (40.6%)	0.812
Net personal income (monthly)				
<500 €	35 (20.6%)	26 (19.5%)	8 (25.0%)	
500–999 €	90 (52.9%)	70 (52.6%)	19 (59.4%)	
1000–1499 €	20 (11.8%)	15 (11.3%)	5 (15.6%)	
1500–1999 €	12 (7.1%)	10 (7.5%)	0	
2000–2499 €	6 (3.5%)	6 (4.5%)	0	
2500–2999 €	2 (1.2%)	2 (1.5%)	0	
not specified	5 (2.9%)	4 (3.0%)	0	0.392
Education (school based)				
<12 years	53 (31.2%)	37 (27.8%)	11 (35.5%)	
≥12 years	113 (66.5%)	94 (70.7%)	19 (61.3%)	
not specified	4 (2.3%)	2 (1.5%)	2 (6.2%)	0.54
Educational level (CASMIN)				
Low	34 (24.3%)	25 (22.9%)	7 (26.9%)	
Middle	58 (41.4%)	44 (40.4%)	14 (53.8%)	
High	48 (34.3%)	40 (36.7%)	5 (19.2%)	0.229
Work permit				
Yes	151 (88.8%)	117 (88.0%)	29 (90.6%)	
No	12 (7.1%)	11 (8.3%)	1 (3.1%)	
Not specified	10 (5.9%)	5 (3.7%)	2 (6.2%)	0.512
Status of employment ^2^				
Not employed	117 (68.8%)	90 (67.7%)	24 (75.0%)	
Marginally employed	24 (14.1%)	20 (15.0%)	4 (12.5%)	
Part-time employment	15 (8.8 %)	13 (9.8%)	2 (6.2%)	
Full-time employment	12 (7.1%)	8 (6.0%)	2 (6.2%)	
Not specified	2 (1.2%)	2 (1.5%)	/	0.886
Status of unemployment ^2^				
Integration program	7 (5.7%)	6 (6.2%)	1 (4.3%)	
Federal Voluntary Service	5 (4.1%)	4 (4.1%)	1 (4.3%)	
Apprentice	4 (3.2%)	4 (4.1%)	/	
Labour market (re-)training	1 (0.8%)	1 (1.0%)	/	
Language course	16 (13.0%)	11 (11.3%)	5 (21.7%)	
Solely housekeeping	14 (11.4%)	13 (13.4%)	1 (4.3%)	
Student	43 (35.0%)	33 (34.0%)	8 (34.8%)	
Registered as unemployed	27 (22.0%)	23 (23.7%)	4 (17.4%)	
Not specified	6 (4.9%)	2 (2.1%)	3 (13.0%)	0.278
Living situation				
Alone	38 (22.3%)	33 (24.8%)	4 (12.5%)	
With family/relatives	98 (57.6%)	73 (54.9%)	22 (68.7%)	
With other people (private home)	31 (18.2%)	25 (18.8%)	5 (15.6%)	
Communal accommodation	3 (1.8%)	2 (1.5%)	1 (3.1%)	0.373
Family status				
Single	83 (48.8%)	69 (51.9%)	11 (34.4%)	
Married	69 (40.6%)	51 (38.3%)	16 (50.0%)	
Divorced	10 (5.9%)	7 (5.3%)	3 (9.4%)	
Widowed	3 (1.8%)	3 (2.3%)	0	
Not specified	5 (2.9%)	3 (2.3%)	2 (6.2%)	0.259
Residence Status ^3^				
Asylum applicant	21 (12.3%)	15 (11.3%)	4 (12.5%)	
Residence permit:				
Refugee status	74 (43.5%)	63 (47.4%)	10 (31.2%)	
Subsidiary protection	14 (8.2%)	11 (8.3%)	3 (9.4%)	
Humanitarian protection	40 (23.5%)	27 (20.3%)	12 (37.5%)	
Not specified	21 (12.3%)	17 (12.8%)	3 (9.4%)	0.408

^1^ randomized = eligible for study participation, study group allocation was performed; nonrandomized: prospect was not eligible for study participation after screening for inclusion and exclusion criteria; ^2^ multiple answers possible, such as “marginally employed” and “student”; ^3^ according to the Federal Office for Migration and Refugees [29], asylum seekers have successfully completed the application procedures, asylum applicants are still in the process, being provided temporary residence status. Ø = mean score.

**Table 2 ijerph-17-07578-t002:** Escape- and trauma-related characteristics of the recruitment sample screened for eligibility in the “Sanadak” trial with regard to randomization status.

Variables	Recruitment Sample (*n* = 170)	Randomized ^1^ (*n* = 133)	Non-Randomized ^1^ (*n* = 32)	*p*-Value
Escape route ^2^				
By airplane	76 (44.7%)	59 (44.4%)	15 (46.9%)	
Via land	113 (66.5%)	86 (64.7%)	24 (75.0%)	
Via sea	99 (58.2%)	76 (57.1%)	20 (62.5%)	
Via transit country	31 (18.2%)	23 (17.3%)	7 (21.9%)	0.985
Time since Syria was left (month)	Ø 49.8 (18.7)	Ø 50.4 (19.3)	Ø 47.2 (16.9)	0.39
Time spent in Germany (month)	Ø 41.0 (13.3)	Ø 41.2 (13.9)	Ø 38.9 (10.8)	0.383
Traumatic event (PDS-5)				
Serious, life-threatening illness	7 (4.1%)	5 (3.8%)	2 (6.2%)	
Physical assault	11 (6.5%)	9 (6.8%)	1 (3.1%)	
Sexual assault	2 (1.2%)	0	2 (6.2%)	
Military/combat-related	56 (32.9%)	47 (35.3%)	7 (21.9%)	
Child abuse	3 (1.8%)	3 (2.3%)	0	
Accident	4 (2.3%)	3 (2.3%)	1 (3.1%)	
Torture/Imprisonment	14 (8.2%)	10 (7.5%)	4 (12.5%)	
Other ^3^	66 (38.8%)	54 (40.6%)	11 (34.4%)	
Not further specified	7 (4.1%)	2 (1.5%)	4 (12.5%)	0.007, V = 0.357
Posttraumatic stress symptoms (PDS-5) ^2^	Ø 23.6 (13.2)	Ø 24.4 (11.1)	Ø 20.3 (20.5) 1	0.141
Intrusion	5.5 (4.1)	5.6 (3.8)	4.5 (5.1)	0.177
Avoidance	2.6 (2.2)	2.6 (2.2)	2.6 (2.4)	0.883
Change in Cognition & Mood	8.5 (5.9)	8.7 (5.4)	7.4 (8.1)	0.297
Arousal & Hyperactivity	7.1 (4.4)	7.4 (3.8)	6.0 (6.5)	0.11

Note: Ø = mean score; PDS-5 = Posttraumatic Diagnostic Scale for DSM-5; V = Cramér’s V effect size; ^1^ randomized = eligible for study participation, study group allocation was performed; nonrandomized: prospect was not eligible for study participation after screening for inclusion and exclusion criteria; ^2^ missing data: *n* = 3, not included; ^3^ in most cases specified as escape-related events.

**Table 3 ijerph-17-07578-t003:** Secondary mental health characteristics of the recruitment sample screened for eligibility in the “Sanadak” trial with regard to randomization status.

Variables	Recruitment Sample	Randomized ^1^ (*n* = 133)	Non-Randomized ^1^ (*n* = 32)	*p*-Value
	(*n* = 170)			
Depressive symptoms (PHQ-9)	Ø 9.4 (5.8)	Ø 9.4 (5.1)	Ø 9.5 (8.2)	0.956
No/low symptom severity (<10)	91 (53.5%)	70 (52.6%)	19 (59.4%)	
Clinical significant severity(≥10)	79 (46.5%)	63 (47.4%)	13 (40.6%)	0.492
Resilience (RS-13)	Ø 64.3 (12.7)	Ø 64.4 (11.4)	Ø 63.1 (17.0)	0.615
Low (13–66)	88 (51.8%)	70 (52.6%)	16 (50.0%)	
Middle (67–72)	32 (18.8%)	25 (18.8%)	6 (18.7%)	
High (73–91)	50 (29.4%)	38 (28.6%)	10 (31.2%)	0.952
Suicidal risk (DSI-SS)	Ø 0.4 (1.3)	Ø 0.2 (1.0)	Ø 1.0 (2.1)	0.002,
				d = −0.623
No suicidal risk (<3)	160 (94.1%)	128 (96.2%)	26 (81.2%)	0.002,
Elevated suicidal risk (≥3)	10 (5.9%)	5 (3.8%)	6 (18.7%)	V = 0.238

Note: Ø = mean score; PHQ-9 = Patient Health Questionnaire; RS-13 = Resilience Scale; DSI-SS = Depressive Symptom Inventory-Suicidality Subscale, d = Cohen’s d effect size; V = Cramér’s V effect size; ^1^ randomized = eligible for study participation, study group allocation was performed; nonrandomized: prospect was not eligible for study participation after screening for inclusion and exclusion criteria.

**Table 4 ijerph-17-07578-t004:** Sociodemographic characteristics of the “Sanadak” study sample with regard to group allocation.

Variables	Study Sample	Intervention Group (*n* = 65)	Control Group (*n* = 68)	*p*-Value
	(*n* = 133)			
Age (M, SD)	Ø 33.3 (11.2)	Ø 33.0 (11.0)	Ø 33.7 (11.4)	0.723
Gender				
Male	82 (61.6%)	43 (66.2%)	39 (57.4%)	
Female	51 (38.3%)	22 (33.8%)	29 (42.6%)	0.297
Net personal income				
<500 €	26 (19.5%)	13 (20.0%)	13 (19.1%)	
500–999 €	70 (52.6%)	32 (49.2%)	38 (55.9%)	
1000–1499 €	15 (11.3%)	7 (10.8%)	8 (11.8%)	
1500–1999 €	10 (7.5%)	6 (9.2%)	4 (5.9%)	
2000–2499 €	6 (4.5%)	5 (7.7%)	1 (1.5%)	
2500–2999 €	2 (1.5%)	0	2 (2.9%)	
not specified	4 (3.0%)	2 (3.1%)	2 (2.9%)	0.472
Education (school-based)				
<12 years	37 (27.8%)	16 (24.6%)	21 (30.9%)	
≥12 years	94 (70.7%)	47 (72.3%)	47 (69.1%)	
No school visit	2 (1.5%)	2 (3.1%)	0	0.271
Educational level (CASMIN)				
Low	25 (22.9%)	12 (24.0%)	13 (22.0%)	
Middle	44 (40.4%)	18 (36.0%)	26 (44.1%)	
High	40 (36.7%)	20 (40.0%)	20 (33.9%)	0.685
Work permit				
Yes	117 (88.0%)	57 (87.7%)	60 (88.2%)	
No	11 (8.3%)	6 (9.2%)	5 (7.3%)	
Not specified	5 (3.7%)	2 (3.1%)	3 (4.4%)	0.933
Status of employment ^1^				
Not employed	90 (67.7%)	40 (61.5%)	50 (73.5%)	
Marginally employed	20 (15.0%)	10 (15.4%)	10 (14.7%)	
Part-time employment	13 (9.8%)	6 (9.2%)	7 (10.3%)	
Full-time employment	8 (6.0%)	8 (12.3%)	/	
Not specified	2 (1.5%)	1 (1.5%)	1 (1.5%)	0.058
Status of unemployment ^1^				
Integration program	6 (6.2%)	2 (4.8%)	4 (7.1%)	
Federal Voluntary Service	4 (4.1%)	1 (2.4%)	3 (5.4%)	
Apprentice	4 (4.1%)	3 (7.1%)	1 (1.8%)	
Labour market (re-)training	1 (1.0%)	/	1 (1.8%)	
Language course	11 (11.3%)	7 (16.7%)	4 (7.1%)	
Solely housekeeping	13 (13.4%)	6 (14.3%)	7 (12.5%)	
Student	33 (34.0%)	12 (28.6%)	21 (37.5%)	
Registered as unemployed	23 (23.7%)	8 (19.0%)	15 (26.8%)	
Not specified	3 (3.1%)	3 (7.1%)	/	0.322
Living situation				
Alone	33 (24.8%)	18 (27.7%)	15 (22.1%)	
With family/relatives	73 (54.9%)	35 (53.8%)	38 (55.9%)	
With other people (private home)	25 (18.8%)	12 (18.5%)	13 (19.1%)	
Communal accommodation	2 (1.5%)	0	2 (2.9%)	0.718
Family status				
Single	69 (51.9%)	37 (56.9%)	32 (47.1%)	
Married	51 (38.3%)	21 (32.3%)	30 (44.1%)	
Divorced	7 (5.3%)	4 (6.1%)	3 (4.4%)	
Widowed	3 (2.3%)	2 (3.1%)	1 (1.5%)	
Not specified	3 (2.3%)	1 (1.5%)	2 (2.9%)	0.61
Residence status ^2^				
Asylum applicant	15 (11.3%)	7 (10.8)	8 (11.8%)	
Residence permit				
Refugee status	63 (47.4%)	31 (47.7%)	32 (47.0%)	
Subsidiary protection	11 (8.3%)	5 (7.7%)	6 (8.8%)	
Humanitarian protection	27 (20.3%)	12 (18.5%)	15 (22.1%)	
Not specified	17 (12.8%)	10 (15.4%)	7 (10.3%)	0.945
Religion and religiosity				
Religious group				
Muslim	74 (56.9%)	38 (56.9%)	36 (54.5%)	
Sunnis	51 (68.9%)	26 (68.4%)	25 (69.4%)	
Shiites	2 (2.7%)	1 (2.6%)	1 (2.8%)	
Alawis	4 (5.4%)	3 (7.9%)	1 (2.8%)	
Not further specified	17 (23.0%)	8 (21.1%)	9 (25.0%)	
Christian	9 (6.9%)	6 (9.4%)	3 (4.5%)	
Other	12 (9.2%)	6 (9.4%)	6 (9.4%)	
None	35 (26.9%)	14 (10.8%)	21 (16.2%)	0.71
Religiosity	Ø 18.50 (4.17)	Ø 18.51 (3.89)	Ø 18.49 (4.46)	0.983
Not religious (<10)	0 (0%)	0 (0%)	0 (0%)	
Religious (10–19)	68 (52.3%)	38 (58.5%)	30 (46.2%)	
Very religious (>19)	62 (47.7%)	27 (41.5%)	35 (53.8%)	0.16

^1^ Multiple answers possible, such as “marginally employed” and “student”; ^2^ according to the Federal Office for Migration and Refugees [29], asylum seekers have successfully completed the application procedures, asylum applicants are still in the process, being provided temporary residence status. Ø = mean score.

**Table 5 ijerph-17-07578-t005:** Escape- and trauma-related characteristics of the “Sanadak” study sample with regard to group allocation.

Variables	Study Sample	Intervention Group (*n* = 65)	Control Group	*p*-Value
	(*n* = 133)		(*n* = 68)	
Escape route ^1^				
By airplane	59 (44.4%)	25 (38.5%)	34 (50.0%)	
By land	86 (64.7%)	43 (66.1%)	43 (63.2%)	
By sea	76 (57.1%)	37 (56.9%)	39 (57.3%)	
Via transit country	23 (17.3%)	13 (20.0%)	10 (14.7%)	0.669
Years since Syria was left (yrs)	Ø 50.4 (19.3)	Ø 3.9 (1.6)	Ø 3.7 (1.6)	0.472
Time spent in Germany (yrs)	Ø 41.2 (13.9)	Ø 3.0 (1.1)	Ø 3.0 (1.3)	1
Traumatic event (PDS-5)				
Serious, life-threatening illness	5 (3.8%)	4 (6.1%)	1 (1.5%)	
Physical assault	9 (6.8%)	4 (6.1%)	5 (7.3%)	
Military/combat-related	47 (35.3%)	21 (32.3%)	26 (38.2%)	
Child abuse	3 (2.3%)	1 (1.5%)	2 (2.9%)	
Accident	3 (2.3%)	2 (3.1%)	1 (1.5%)	
Torture/imprisonment	10 (7.5%)	3 (4.6%)	7 (10.3%)	
Other	54 (40.6%)	29 (44.6%)	25 (36.8%)	
Refusal of answer	2 (1.5%)	1 (1.5%)	1 (1.5%)	0.667
Post-traumatic stress symptoms (PDS-5)	Ø 23.8 (11.6)	Ø 23.2 (10.8)	Ø 24.4 (12.4)	0.539
Intrusion	5.6 (4.1)	5.4 (4.0)	5.9 (4.2)	0.483
Avoidance	2.6 (2.1)	2.5 (1.9)	2.7 (2.2)	0.584
Change in cognition & mood	8.4 (5.4)	8.0 (5.1)	8.8 (5.8)	0.439
Arousal & hyperactivity	7.1 (3.7)	7.2 (3.8)	7.0 (3.6)	0.772

Note: Ø = mean score; PDS-5 = Posttraumatic Diagnostic Scale for DSM-5.

**Table 6 ijerph-17-07578-t006:** Secondary characteristics of the “Sanadak” study sample with regard to group allocation.

Variables	Study Sample	Intervention Group (*n* = 65)	Control Group (*n* = 68)	*p*-Value
	(*n* = 133)			
Depressive symptoms (PHQ-9)	Ø 9.2 (5.2)	Ø 9.2 (4.8)	Ø 9.3 (5.7)	0.84
No/low symptom severity (<10)	73 (54.9%)	37 (56.9%)	36 (52.9%)	
Clinical significant severity(≥10)	60 (45.1%)	28 (43.1%)	32 (47.1%)	0.645
Generalized anxiety (GAD-7)	Ø 8.5 (5.0)	Ø 8.2 (4.4)	Ø 8.8 (5.5)	0.483
No/low symptom severity (<10)	79 (59.4%)	40 (61.5%)	39 (57.3%)	
Clinical significant severity(≥10)	54 (40.6%)	25 (38.5%)	29 (42.6%)	0.623
Somatization (PHQ-15)				
Female	Ø 10.5 (5.3)	Ø 10.0 (5.2)	Ø 11.1 (5.4)	0.441
Low symptom severity (<10)	18 (35.3%)	12 (41.4%)	6 (27.3%)	
Medium-high symptom severity (≥10)	33 (64.7%)	17 (58.6%)	16 (72.7%)	0.296
Male	Ø 7.8 (5.1)	Ø 7.7 (5.0)	Ø 7.9 (5.3)	0.853
Low symptom severity (<10)	52 (63.4%)	22 (56.4%)	30 (69.8%)	
Medium-high symptom severity (≥10)	30 (36.6%)	17 (43.6%)	13 (30.2%)	0.21
Social network size (LSNS-6)	Ø 15.1 (5.3)	Ø 15.0 (5.5)	Ø 15.2 (5.2)	0.785
Social isolation (LSNS-6 < 12)	43 (32.3 %)	21 (32.3%)	22 (32.4%)	0.996
Social support (ESSI)	Ø 18.0 (4.7)	Ø 18.4 (4.1)	Ø 17.7 (5.2)	0.388
Low support	71 (53.4%)	32 (49.2%)	39 (57.4%)	
High support	62 (46.6%)	33 (50.8%)	29 (42.6%)	0.348
General self-efficacy (GSE)	Ø 27.4 (4.7)	Ø 26.8 (5.2)	Ø 28.0 (4.0)	0.151
Resilience (RS-13)	Ø 64.4 (11.4)	Ø 61.6 (11.6)	Ø 67.0 (10.3)	0.006,
				d = 0.480
Low (13–66)	70 (52.6%)	41 (63.1%)	29 (42.6%)	
Middle (67–72)	25 (18.8%)	11 (16.9%)	14 (20.6%)	0.046,
High (73–91)	38 (28.6%)	13 (20.0%)	25 (36.8%)	V = 0.215
Self-stigmatization (SSMIS-SF)				
Awareness	28.3 (7.5)	28.8 (8.3)	27.8 (6.6)	0.449
Agreement	18.8 (6.7)	17.6 (6.5)	20.0 (6.8)	0.034, d = −0.372
Application	16.5 (6.6)	15.2 (6.2)	17.8 (6.8)	0.021, d = −0.404
Harm to Self-esteem	18.6 (9.5)	18.0 (9.5)	19.2 (9.6)	0.454
Health-related quality of life & subjective health				
EQ-5D-5L	0.82 (0.19)	0.79 (0.23)	0.86 (0.13)	0.052
EQ-VAS	73.6 (18.9)	73.0 (20.7)	74.2 (16.9)	0.713
Suicidal risk (DSI-SS)	Ø 0.2 (1.0)	Ø 0.03 (0.2)	Ø 0.2 (1.1)	0.168
<3 (no suicidal risk)	128 (96.2%)	64 (98.5%)	65 (95.6%)	
≥3 (elevated suicidal risk)	5 (3.8%)	1 (1.5%)	3 (4.4%)	0.268

Note: Ø = mean score; DSI-SS = Depressive Symptom Inventory-Suicidality Subscale; EQ-5D_5L = 5-level version of EQ-5D; EQ-VAS = EQ visual analogue scale; ESSI = ENRICHD Social Support Inventory; GAD-7 = Generalized Anxiety Disorder Scale-7; GSE = general self-efficacy; LSNS-6 = short form of the Lubben Social Network Scale; PHQ-9/-15 = Patient Health Questionnaire; RS-13 = Resilience Scale; SSMIS-SF = Self-Stigma of Mental Illness Scale–Short Form; d = Cohen’s d effect size; V = Cramer’s V effect size.

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
