# Peer review of "Recruitment and Baseline Characteristics of Participants in the “Sanadak” Trial: A Self-Help App for Syrian Refugees with Post-traumatic Stress"

_ijerph, 2020, doi:10.3390/ijerph17207578_

Round 1

Reviewer 1 Report

Thank you very much for this innovative intervention (self-help app) and which is a research project with several steps  before evaluating the intervention with trial and control sample. Further, how will they perceive an app compared to traditional therapy in a room with or without an interpreter? Especially when they come from a relational cultural background? Can the self-help app be spread by collaboration in other EU-countries?

Would it be possible also to use the self-help app for dari, kurdish, somalian refugees? 

How to use the self-help app when you can not read and right?

What is the ethical identity number?

Author Response

C1: Thank you very much for this innovative intervention (self-help app) and which is a research project with several steps  before evaluating the intervention with trial and control sample.

We would like to thank the reviewer for their time and effort taken to evaluate our work.

C2: Further, how will they perceive an app compared to traditional therapy in a room with or without an interpreter? Especially when they come from a relational cultural background?

Good point. The app is designed as a low-threshold self-help in Arabic language and culturally sensitive to the Syrian cultural background - purposely to work without the support of an interpreter. Our trial evaluates the app against a control group that receives psychoeducation reading material. As such, we cannot establish the effectiveness against a face-to-face therapy this time.

C3: Can the self-help app be spread by collaboration in other EU-countries? Would it be possible also to use the self-help app for dari, kurdish, somalian refugees? 

Yes, absolutely, there is potential for collaboration in other EU and also non-EU countries. The app will be made available for free and without barriers after the evaluation. It is possible to translate the app into different languages for potential use among refugees from other regions/countries. The app content is largely created in a way that it addresses general refugee themes associated with escape and post-migration stressors. However, a careful consideration of adaptation of some narrative details (e.g. names and visuals of figures in animated videos) would be necessary. This aspect will be discussed in greater detail in the trial outcome paper once we know whether or not the intervention is effective.

C4: How to use the self-help app when you can not read and right?

The app requires literacy. Besides audios and videos that would work for illiterate individuals, most of the information and exercises provided require the ability to read and write. The usability of the app would be very low for individuals without these abilities.

Changes made in 2.1. Recruitment and recruitment procedures:

Moreover, literacy was a requirement.

C5: What is the ethical identity number?

Good point. We have added the following statement.

Changes made:

 2.5. Ethics

The study was approved by the Ethics committee of the Medical Faculty of the University of Leipzig, Germany (ID: 111–17-ek) and adheres to the Declaration of Helsinki and the ICH guidelines for Good Clinical Practice (GCP). All participants were informed about the study aims, including clarification about data security according to latest legal standards. Participation was only allowed after written informed consent.

Reviewer 2 Report

This is a very well done article providing a part of the larger study information. The larger study focuses on a very important area of research for public mental health interventions through finding innovative means for reaching underserved and targeted populations. 

For the specific focus of this article all is done correctly and presented in a readable and well organized manner.  If one is following the trajectory of the publications for the study in English language presentations, this fits well with the information presented in publication ref. 13. 

The language of the article is good, though a minor language review is needed for adjusting some adjective/adverb to subject formats, removing duplication of words, and some awkward phrasing. Also, some adjustments for alignment in the tables is needed.

There is quite a bit of useful information in this article related to the involvment of the targeted population (language, some cultural concerns, etc.) in the study's design.  However, it would be very helpful in the discussion section at least to know why other potential variables and available measures or questions that might have been anticipated have not been included.  Two seem especially important to mention: perceived as well as experienced discrimination, and religiosity (both as a demographic item in terms of Christian/Muslim identification in consideration of the German context, as well as assessing degree of religiosity (understood broadly) as contributing to possible positive or negative coping strategies.  These are variables that are important for certain migrant groups with respect to the groups themselves (for this group well documented) as well as in consideration of the cultural context in which they find thmeselves. Perhaps some dimension of these variables might be included in some part of the analysis to come. If the case, that would be important to note, or why the intentional exclusion of such. In particular it would have been interesting to see possible correlations with resilience and trauma scores. 

There is some information provided in the article about the information that the control group will receive. As the intervention group it seems will work with both a better understanding of trauma AND better understanding as well as activities for reducing stress and improving wellbeing, is the control group also given information on both trauma AND stress reduction and wellbeing?  A bit more information would be helpful also in this article.

Author Response

C1: This is a very well done article providing a part of the larger study information. The larger study focuses on a very important area of research for public mental health interventions through finding innovative means for reaching underserved and targeted populations. For the specific focus of this article all is done correctly and presented in a readable and well organized manner.  If one is following the trajectory of the publications for the study in English language presentations, this fits well with the information presented in publication ref. 13. 

We really appreciate the reviewer’s evaluation of our manuscript. Many thanks!

C2: The language of the article is good, though a minor language review is needed for adjusting some adjective/adverb to subject formats, removing duplication of words, and some awkward phrasing. Also, some adjustments for alignment in the tables is needed.

We carefully checked the whole manuscript for errors and had an English native speaker do a final edit.

C3: There is quite a bit of useful information in this article related to the involvement of the targeted population (language, some cultural concerns, etc.) in the study's design.  However, it would be very helpful in the discussion section at least to know why other potential variables and available measures or questions that might have been anticipated have not been included.  Two seem especially important to mention: perceived as well as experienced discrimination, and religiosity (both as a demographic item in terms of Christian/Muslim identification in consideration of the German context, as well as assessing degree of religiosity (understood broadly) as contributing to possible positive or negative coping strategies.  These are variables that are important for certain migrant groups with respect to the groups themselves (for this group well documented) as well as in consideration of the cultural context in which they find themselves. Perhaps some dimension of these variables might be included in some part of the analysis to come. If the case, that would be important to note, or why the intentional exclusion of such. In particular it would have been interesting to see possible correlations with resilience and trauma scores. 

Good point. We agree that religiosity is a relevant factors in the context of this study. In fact, we did assess religious group and religiosity (using the Centrality of Religiosity Scale/CSR). We have added this information as well as baseline results of the CSR to the respective section and table (Table 4). Religiosity will be considered in the trial outcome analysis as we are also interested in potential associations with trial outcomes. Baseline assessments were very comprehensive, taking 2.5 to 3 hours on average. We were not able to consider every potentially informative factor, discrimination was not assessed, for example. It was vastly covered as a potential outcome in the trauma assessment, if named as a post-migration stressor. We agree that experienced discrimination is another relevant factor to consider, but again: there were so many factors and in the end, we were not able to cover everything. We have added this consideration to the limitations section.

Changes made:

Please see Table 4.

Lastly, even though assessments were very comprehensive (average duration: 2.5 to 3 hours), there may be other relevant factors, for example perceived discrimination in the host country, that we were not able to cover due to time constraints.

C4: There is some information provided in the article about the information that the control group will receive. As the intervention group it seems will work with both a better understanding of trauma AND better understanding as well as activities for reducing stress and improving wellbeing, is the control group also given information on both trauma AND stress reduction and wellbeing?  A bit more information would be helpful also in this article.

The control group received psychoeducational reading material to increase knowledge about trauma/PTSD/posttraumatic stress only. This did not include specific information on how to reduce stress or improving wellbeing (this aspect was exclusive to the intervention group).

Changes made:

The content of the “Sanadak” app is multi-modal, i.e. it includes psychoeducational information in order to increase knowledge and awareness of PTSD and related mental health issues, and self-help techniques as well as skills training with respect to symptom management. In addition, a short self-test on posttraumatic symptom severity is implemented to allow for automated tailored feedback regarding progress at any time. Interactive materials, such as animated videos and audios as well as games and exercises are provided to maximize usability. The effectiveness of the app is evaluated in a randomized-controlled trial (RCT) with two arms: the intervention group uses the app and the control group receives psychoeducational reading material in order to increase knowledge and awareness of PTSD. Details have been described elsewhere [13].